# A socially interdependent choice framework for social influences in healthcare decision-making: a study protocol

Sven P H Nouwens [1,2,3] Jorien Veldwijk [1,2,3] Luis Pilli [1,2,3] Joffre D Swait [1,2,3] Joanna Coast [4] Esther W de Bekker-Grob [1,2,3]

¹Erasmus Universiteit Rotterdam Erasmus School of Health Policy and Management, Rotterdam, Netherlands
²Erasmus Choice Modeling Centre, Erasmus University Rotterdam, Rotterdam, Netherlands
³Erasmus Centre for Health Economics, Rotterdam, The Netherlands
⁴Population Health Sciences, University of Bristol, Bristol, UK

**Correspondence to**
Sven P H Nouwens;
nouwens@eshpm.eur.nl

## ABSTRACT

**Objectives** Current choice models in healthcare (and beyond) can provide suboptimal predictions of healthcare users' decisions. One reason for such inaccuracy is that standard microeconomic theory assumes that decisions of healthcare users are made in a social vacuum. Healthcare choices, however, can in fact be (entirely) socially determined. To achieve more accurate choice predictions within healthcare and therefore better policy decisions, the social influences that affect healthcare user decision-making need to be identified and explicitly integrated into choice models. The purpose of this study is to develop a socially interdependent choice framework of healthcare user decision-making.

**Design** A mixed-methods approach will be used. A systematic literature review will be conducted that identifies the social influences on healthcare user decision-making. Based on the outcomes of a systematic literature review, an interview guide will be developed that assesses which, and how, social influences affect healthcare user decision-making in four different medical fields. This guide will be used during two exploratory focus groups to assess the engagement of participants and clarity of questions and probes. The refined interview guide will be used to conduct the semistructured interviews with healthcare professionals and users. These interviews will explore in detail which, and how, social influences affect healthcare user decision-making. Focus group and interview transcripts will be analysed iteratively using a constant comparative approach based on a mix of inductive and deductive coding. Based on the outcomes, a social influence independent choice framework for healthcare user decision-making will be drafted. Finally, the Delphi technique will be employed to achieve consensus about the final version of this choice framework.

**Ethics and dissemination** This study was approved by the Erasmus School of Health Policy and Management Research Ethics Review Committee (ESHPM, Rotterdam, The Netherlands; reference ETH2122-0666).

## INTRODUCTION

Rising expenditures, an ageing population, high prices of new medical treatment options and substantial annual resource waste put

---

## STRENGTHS AND LIMITATIONS OF THIS STUDY

⇒ This is the first study that will attempt to develop a socially interdependent choice framework of healthcare user decision-making.
⇒ Through applying a mixed-method approach, existing knowledge and empirical evidence will be combined to ensure a rigorous approach for framework development.
⇒ Inclusion of articles based on the study's title and abstract in the systematic literature review step may result in potentially relevant areas remaining unexamined. In part, this will be corrected in the interviews and Delphi method steps, which allow influences and influence mechanisms to surface later in the study.
⇒ Participant recruitment via healthcare providers may lead to selection bias. However, recruited participants will likely be experienced with healthcare decision-making within a medical field of interest.

---

healthcare under heavy pressure.[1–4] To avoid poor policy decisions, trial-and-error implementation and demand-supply imbalance, there is a need for accurate predictions of healthcare users' choice behaviour.[5 6] Current choice models in healthcare (and beyond), however, provide inaccurate predictions of individual healthcare user decisions. One reason for such inaccuracy is that standard microeconomic theory assumes that decision-making occurs in a social vacuum. Choice behaviour, under this homo economicus perspective,[7] is regarded as a completely rational process in which the decision-maker is driven purely by their own interest, disregarding any self-originating or other-originating psychological factors such as social influences. However, healthcare user choices can in fact be (entirely) socially determined.[8–12] To achieve more accurate choice behaviour predictions within healthcare, and therefore, better policy decisions, the social

influences that affect healthcare users' decision-making need to be identified and explicitly integrated into choice models.

This motivates the need for an expanded choice paradigm that must be developed and validated. In the pursuit of this objective, a first important step is to develop and extend the theory of decision-making by building a conceptual framework that depicts which social influences impact healthcare user decision-making, and how they achieve their impact. With this in mind, we developed a systematic approach that (1) empirically identifies these social influences and (2) suggests mechanisms for their impacts. We strive towards these objectives using a phased approach: a systematic literature review, a series of focus groups and semistructured interviews and the Delphi method will supplement each other. This paper provides the protocol for the systematic approach that will be taken towards developing this socially interdependent choice framework.

## OBJECTIVES

This study is part of the INTERSOCIAL project that aims to develop and validate a socially interdependent choice paradigm for healthcare user decision-making such that more accurate predictions of choice behaviour in healthcare can be achieved.[13] This study protocol will focus on conducting the necessary steps that will guide the attempt to extend the existing theoretical choice framework to account for social influences. The objectives for this study protocol are as follows:

1. To identify which, and how, social influences have previously been found to affect healthcare user decision-making based on existing published peer-reviewed studies.
2. To explore with healthcare users in four different areas which, and how, social influences affect healthcare user decision-making.
3. To develop consensus among experts (including healthcare professionals) about which, and how, social influences affect healthcare user decision-making.

## DEFINITION OF SOCIAL INFLUENCE

There exist many definitions of social influence[14–17] with different levels of practical applicability. Based on a consensus meeting with the research team and external experts (n=16) from different academic fields, it was decided to use the following definition in the present study: 'social influence is the process by which an individual's behaviour, thoughts, beliefs, attitudes, feelings, actions, preferences, decisions or choice behaviour are (directly and/or indirectly) affected by other people'. These 'other people' can be anyone that the decision maker (in our case, users of healthcare) is in direct contact with but also individuals that the decision-maker is not in direct contact with, for example, celebrity endorsements.[18]

Note that our definition is centred on a decision-maker whose healthcare decisions may depend on other people in the decision-maker's social circle, implying that we are excluding the study of networks of individual healthcare users who are jointly influenced by each other.[19] We apply a unidirectional paradigm where the entire social network is centred on a single healthcare user and study how this social network ((types of) individuals and processes) influences the healthcare user, but not how the healthcare user influences others' social networks or how a healthcare user's social contacts are influenced by each other. We only include social influences that can be tracked to a single source. Other social influences (such as culture,[20] education[21] and perceived discrimination[22]) that cannot be tracked to a single source will be documented only.

## RESEARCH DESIGN

The research process will use a mixed-methods approach. First, a systematic literature review that identifies the (types of) individuals and the processes that systematically influence healthcare users' decisions within different medical fields will be performed. The results from the systematic literature review will aid in constructing an interview guide for the subsequent round of (group and individual) interviews. As individuals are often unconscious and unaware of the degree to which they are socially influenced,[23 24] the results from the systematic literature review can additionally help to construct interview questions that stimulate recall and awareness of social influences that might otherwise not be raised by respondents. Second, 2 focus groups and 40 semistructured interviews will be conducted to explore and assess healthcare users' and healthcare professionals' views on social influences on healthcare user decision-making. The focus groups have a secondary aim of testing engagement of participants with the topic and assessing the clarity of questions and probes in the interview guide. Their feedback will then be used to refine the interview guide for the semistructured interviews. Third, the Delphi method will be applied to reach consensus among experts (including healthcare professionals and choice modellers) about the socially interdependent choice framework on healthcare user decision-making.

For the systematic literature review, the following medical fields of study have been chosen: vaccination, birth setting, arthroplasty and prostate cancer treatment. The rationale for choosing these fields is fourfold. First, healthcare users in these fields face a degree of autonomy in their choice, meaning that they are not obliged to choose one alternative over the other, which is essential for testing real choice behaviour predictability. Second, many healthcare users face these medical fields (in the Netherlands, which has 17.5 million inhabitants: almost all people are affected by vaccinations, between 160 000 and 180 000 people give birth yearly,[25] 31 514 hip replacements were reported in 2022[26] and prevalence of prostate

cancer is over 55 000[27]) speaking to relevance from the perspective of development of an extended socially interdependent framework. Third, a higher degree of generalisability is achieved by including medical fields that are likely to differ in types/degrees of social interdependence and consequences that may employ different mechanisms of social influence. Fourth, in each of these four fields, there are current policy challenges related to the decisions made by healthcare users in these areas: (1) the demand for influenza vaccination is lower than for other vaccinations,[28] (2) there is increasing interest in birth centres in the Dutch population,[29] (3) there is an exponential rise in patients with joint problems[30 31] and (4) there is an alarming increase in active-surveillance drop-out from prostate cancer patients.[32 33]

## SYSTEMATIC LITERATURE REVIEW

Initially, a systematic literature review of social influences on healthcare user decision-making will be performed. The review will follow the Preferred Reporting Items for Systematic Reviews and Meta-Analyses(PRISMA) guidelines[34] and map the literature in three dimensions that concern the development of the socially interdependent choice paradigm proposed by the INTERSOCIAL project.[13] These dimensions describe (1) the individuals or groups influencing health-related decision-making; (2) the mechanisms through which they influence decisions of healthcare users and (3) the choice constructs that these processes affect. A narrative synthesis of the sources and mechanisms of social influence and choice constructs affected will be provided.

Concerning the first dimension, psychosocial variables are one of the key facets of patient-centred care.[35] The past literature offers evidence of treatment choices as an action embedded in a network of social relationships, including experts and non-experts.[36 37] This first dimension focuses on which individuals' and groups' actions must be considered for the development of the theoretical framework for healthcare user decision-making.

The second dimension, that is, social influence mechanisms, refers to the processes by which a healthcare user's behaviour, thoughts, beliefs, attitudes, feelings, actions, preferences, decisions or choice behaviours are (directly and/or indirectly) affected by other people.[38 39] Mechanisms and dimensions already present in the health-related literature will support classifying included papers into categories. Physician–patient interaction[36 37] and social support[40] are two relevant and well-defined constructs that may impact healthcare user decision-making. Social norm is a relevant construct from sociocognitive theories, and it is present in many attitudinal-behavioural models in the health domain, such as the Health Belief[41] model, I-Change[42 43] model and the Transtheoretical Model of Behaviour Change.[44]

The creation of a socially interdependent choice framework demands the connection between the first two dimensions and the choice constructs involved in choice econometric models, such as the choice framework of Dellaert et al,[45] which describes how individuals evaluate attributes and make decisions in the presence of multiple goals. Since insights about this connection from the extant literature are sparse at best, choice constructs will be aggregated into three broadly defined main categories. These categories contain the effects of social influences on (1) healthcare users' goals, (2) preference formation and (3) non-evaluative components of choice models. Goals involve crucial personal trade-offs such as life expectancy vs quality of life, where different alternatives are valued differently based on the goal that a healthcare user seeks to attain, with other persons possibly influencing the activation of certain goals. Preference formation describes the effects of social influences on the acquisition of information about attributes and the deployment of weights to these attributes to determine options' total utilities. Finally, non-evaluative components refer to any steps of the decision-making process where certain options are excluded based on single attributes, instead of an option's unique blend of attribute levels. Examples of non-evaluative components are habitual behaviour, non-compensatory processes or screening of alternatives before evaluation.

To attain the objectives of the systematic literature review, nine different databases will be searched: Embase, Cochrane CENTRAL, Econlit, Google Scholar, International Bibliography of the Social Sciences, Medline, PsycINFO, Scopus and Web of Science. To be eligible, a paper must return from a search containing (variations around) the keywords 'social influence' AND 'decision-making' AND 'empirical data' AND 'healthcare field'. The 'healthcare field' differs for each of the four healthcare fields studied. For example, the field of vaccination will use the keywords 'vaccine' OR 'vaccine hesitancy' OR 'vaccination' OR 'vaccination coverage'. We will exclude the terms 'parental' and 'parental decision-making', as we are interested in how a healthcare user makes choices about their own health and how social influences affect these choices.

The inclusion criteria further require that the full text of a paper is accessible through the institutional subscription of any of the project collaborators, is written in English and presents results of original empirical data collected in Western countries. Countries that will be included in the search strategy are the USA, Canada, North America, UK, England, Ireland, Northern Ireland, Scotland, the Netherlands, Belgium, France, Germany, Denmark, Norway, Sweden, Finland, Spain, Portugal, Italy, Austria, Switzerland, Iceland, Luxembourg, Liechtenstein, San Marino, Monaco, Andorra, Malta, Cyprus, Gibraltar, Europe, Australia and New Zealand. Papers reporting data from other countries will be excluded due to the assumption of significantly divergent cultures and social environments compared with Western countries.

The analysis will follow a two-step deductive-inductive approach. First, in the deductive step, we will extract data from papers coding the social influence sources,

mechanisms (such as doctor–patient interaction or social support) and choice constructs using the framework from Dellaert *et al.*[45] From this initial data extraction round, we will streamline the definitions of social influence mechanisms and choice constructs. Based on the streamlined definitions, we will inductively propose the connection of sources and mechanisms of social influence and choice constructs. The last step of the process is to describe frequencies and co-occurrences of relevant individuals and groups, social influence mechanisms and choice constructs, and differences across medical areas.

From a broader perspective, the systematic literature review will support the identification of important social influences, their mechanisms and affected choice constructs in the extant literature. This will allow the subsequent steps to exploit the signals and explore the gaps.

## EXPLORATORY FOCUS GROUPS AND SEMISTRUCTURED INTERVIEWS

Based on the outcomes of the systematic literature review, using best-practice guidelines,[46 47] an interview guide will be developed that assesses which, and how, social influences affect healthcare user decision-making according to healthcare users. Two exploratory focus groups will be conducted to enable the further development of the content within the interview guide through discussion of issues that the research team may not have considered, to test the engagement of participants with the topic, and to assess the clarity of questions and probes in the interview guide (and make any adjustments). One focus group will be conducted with people aged 60 years and older, who were or will be offered influenza vaccination and the second focus group among pregnant women who were or are choosing their birth setting. Each group will consist of 6–10 participants[46] and will be conducted in person.

One important aspect of the focus groups is to test and further generate the interview guide. As this is a complex topic which may not be immediately understandable to participants, attention will be given to whether questions were understood by the participants (ie, not considered too abstract in nature), whether the formulation of the questions stimulated understanding of the topic, and whether probes were clear. The focus groups may also introduce topics that were not expected, and the interview guide will be adjusted to enable these topics to be captured during individual interviews.

Subsequently, semistructured interviews will be conducted. The semistructured interviews will provide a detailed exploration of which, and how, social influences affect healthcare user decision-making. These interviews will be conducted among healthcare users (decision-makers) and healthcare professionals in the medical fields of influenza vaccination, prostate cancer treatment, joint replacement and birth setting. Per medical field, a minimum of 10 individuals will be recruited consisting of healthcare users (n=5) and healthcare professionals

(n=5).[48] Additional interviews will be added based on whether data saturation has been reached within each medical field of interest. Interview guides will be adjusted as the analysis progresses, to take account of new and emerging findings and to pursue new lines of enquiry. Interviews will preferably be face-to-face at a convenient location for the interviewee, however, if requested by the interviewee, these in-depth interviews can be conducted either via a networking facility (eg, Microsoft Teams or Zoom) or telephone.

Participants will be recruited through already confirmed collaborations with Erasmus MC—University Medical Centre clinicians, general practices, obstetric centre (focus groups and interviews) and the Prostate Centre South-West Netherlands (interviews only). Focus group participants will receive a €20 gift voucher as compensation. For the interviews, participants will receive a €10 gift voucher as compensation if they participate in a face-to-face or online interview. Focus groups and interviews will be audiorecorded, then transcribed verbatim.

Before the focus groups and in-person interviews begin, participants will be provided with information about the study and asked to sign an informed consent form. After consent has been obtained, they will be asked to fill in a short (online) survey including questions on their health literacy[49] and their decision-making style.[50] At the end of the interviews, participants will be presented with their survey answers and given the opportunity to revise them if they would like to do so in light of the discussion.

As part of the interviews, hierarchical mapping will be employed to construct a social proximity map.[51] Participants will be asked to place influential individuals on a diagram of concentric circles, with the decision maker (ie, healthcare user) in the middle. The placement of individuals represents the participant's social network, but also the strength of influence of each specific individual.

The analysis of the transcripts from the focus groups will be conducted prior to the interviews and will focus on the issues around the data collection process, as well as providing some preliminary themes. Transcripts from interviews will be analysed iteratively using a grounded theory constant comparative approach[52] based on a mix of inductive and deductive (from the systematic review/initial framework) coding. The constant comparative approach compares each incident in the data with other incidents for similarities and differences and groups conceptual similarities under higher-level descriptive concepts.[52] The analysis will result in a first version of the theoretical framework of social influence on healthcare user decision-making. For data analysis, Atlas.ti will be used. Data collected through survey and recording devices will be stored for at least ten years through SURF Yoda, in line with Erasmus University policy.

## DELPHI METHOD

The findings obtained in the literature review and qualitative interviews will be consolidated by applying the Delphi

method among experts from different medical (influenza vaccination, prostate cancer treatment, joint replacement and birth setting) and scientific (eg, choice modelling in health economics, marketing, transportation, environmental economics) fields. Linstone and Turoff[53] introduced the Delphi method as 'a method for structuring a group communication process so that the process is effective in allowing a group of individuals, as a whole, to deal with a complex problem.' The current study will use this method to further structure and reach consensus on the developed socially interdependent choice framework.

Experts (researchers and clinicians) from the different fields will be proposed parts of the socially interdependent choice framework and asked to provide feedback in several rounds. Experts will be contacted through email and are given a week to reply to a round of the Delphi method. After each round, a summary of the experts' perspectives will be provided to the experts, along with a revised model of social influence. This will allow the experts to revise their earlier comments in the next round of the Delphi study.

Agreement among two-thirds (67%) of participants on a model of social influence will be considered the threshold for consensus.[53] Besides consensus, group stability will also be measured. This gives an indication of consistency across rounds. A set of $\chi^2$ tests will be performed to keep track of group stability.[54] If stability and consensus have both been reached, the procedure will be stopped. If after five rounds, stability or consensus has not been reached, the procedure will also be stopped. In this case, the version of the socially interdependent framework with the most consensus will be accepted, and dissents will be noted.

## PATIENT INVOLVEMENT

Healthcare users and members of the public were not involved in the design, conduct, reporting or dissemination of this protocol. However, experiences of healthcare users will guide the refinement of the interview guide and eventually the creation of a socially interdependent choice framework in the planned study.

## DISCUSSION

This study will focus on identifying which social influences affect healthcare user decision-making, as well as the mechanisms/processes for their impacts. By forming an understanding of social influences in individual decision-making, this study will not only help to identify social influences in healthcare decision-making but also help to extend choice theory by building an evidence-based interdependent choice framework that depicts which, and how, social influences impact healthcare user decision-making. By incorporating these social influences, which can have a big impact on the choice process,[8–12] such a framework has the potential to contribute towards more accurate predictions of healthcare choices, which could

in turn have a positive impact on decreasing poor policy decisions, trial-and-error implementations and demand-supply imbalances.

## ADDRESSING LIMITATIONS

A limitation of the systematic literature review approach is the inclusion criteria of articles based on the study's title and abstract. This may result in missing findings regarding less straightforward mechanisms of social influences or choice constructs, that are not covered by the search strategy. Additionally, minor findings are more likely to be overlooked, as these are often not mentioned in the abstract or title. This may lead to an over-representation of significant results resulting from the literature review. Further validation would be necessary to estimate the effects and their significance more accurately. The current protocol is designed to (partly) correct for this: interviews and the Delphi method allow influences and influence mechanisms that are not found in the systematic literature step to surface later in the study.

Another limitation, related to the explorative focus groups and semistructured interviews, is the recruitment of participants through healthcare providers. While on the one hand this can count as a strength due to recruiting participants who are likely experienced with healthcare decision-making within a medical field of interest, it may also lead to selection bias: these participants may not be representative of the targeted population in terms of experience with the decisions that are studied.

## ETHICS AND DISSEMINATION

This study was approved by the Erasmus School of Health Policy and Management Research Ethics Review Committee (ESHPM, Rotterdam, The Netherlands; reference ETH2122-0666).

All participants will be provided with an information letter and given the opportunity to ask questions on the study. Informed consent will be obtained before their participation. Data will be anonymised by removing names and any other personal information from transcripts and answers during the Delphi method.

Findings will be published in peer-reviewed academic journals and further shared with various audiences such as researchers, healthcare professionals and policymakers through methods such as public presentations and academic conferences. The gathered data and the constructed data set from the systematic literature review can be requested from the corresponding author. The full interview transcripts will not be made available for request as participants have signed a consent form that promises to maintain complete privacy.

**Acknowledgements** The authors wish to thank Angie Fagerlin for fruitful discussion and comments on an earlier version.

**Contributors** EdB-G, JV, JDS, LP and JC: initial conceptualisation and design of the study. All authors further adapted and improved the design of the study. SN: drafted the manuscript. All authors provided critical revision of the manuscript and have read and approved the final manuscript.

**Funding** The project has received funding from the Dutch Research Council (NWO-Talent-Scheme-Vidi-Grant No. 09150171910002). JC is supported by the Wellcome Trust (205384/Z/16/Z). For the purpose of open access, the author has applied a CC BY public copyright licence to any Author Accepted Manuscript version arising from this submission.

**Competing interests** None declared.

**Patient and public involvement** Patients and/or the public were not involved in the design, or conduct, or reporting, or dissemination plans of this research.

**Patient consent for publication** Not applicable.

**Provenance and peer review** Not commissioned; externally peer reviewed.

**Open access** This is an open access article distributed in accordance with the Creative Commons Attribution 4.0 Unported (CC BY 4.0) license, which permits others to copy, redistribute, remix, transform and build upon this work for any purpose, provided the original work is properly cited, a link to the licence is given, and indication of whether changes were made. See: https://creativecommons.org/licenses/by/4.0/.

**ORCID iDs**
Sven P H Nouwens http://orcid.org/0009-0003-4768-8631
Jorien Veldwijk http://orcid.org/0000-0003-4822-5068
Luis Pilli http://orcid.org/0000-0002-4416-6155
Joffre D Swait http://orcid.org/0000-0002-0435-117X
Joanna Coast http://orcid.org/0000-0002-3537-5166
Esther W de Bekker-Grob http://orcid.org/0000-0001-7645-6168

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
