## [Reviewer comments · BMJ Open]

ARTICLE DETAILS

TITLE (PROVISIONAL)	A Socially Interdependent Choice Framework for Social Influences in Healthcare Decision-Making: A Study Protocol
AUTHORS	Nouwens, Sven; Veldwijk, Jorien; Pilli, Luis; Swait, J.D.; Coast, Joanna; de Bekker-Grob, Esther

VERSION 1 – REVIEW

REVIEWER	Stiggelbout, Anne University Medical Center Leiden, Medical Decisionmaking
REVIEW RETURNED	04-Oct-2023

GENERAL COMMENTS	This is a protocol for an interesting and relevant study on choice models. The aim is to achieve more accurate choice predictions by identifying and integrating into these models the social influences that affect healthcare user decision-making. The authors propose to develop a social interdependent choice framework of healthcare user decision-making. I think this will be an interesting and highly relevant study. My main concerns are textual, upon reading I had many questions, calls for clarification. I provide them simply in the order of the text. A bit confusing was to me that in different places different wording is used, and that at the end, the framework seemed to have disappeared. Moreover how exactly the framework can be used eludes me. "Based on the outcomes a social influence independent choice framework" will be drafted (Design section in Abstract). I guess the term social independent is an error for in the rest of the paper it is interdependent? But perhaps socially interdependent? I suggest to use the more commonly term Vaccination rather than Immunization. On page 10, I would have liked to see some more information on the framework of Dellaert, because the framework has such a prominent place in the study, but as a naïve reader I have not gained much idea about what it will look like. Since it is a deductive step, it would help the reader to see this framework. I really like the following sentence about the inductive part.
--

	Is it truly mixed methods, or multiple methods? Not much is said about how studies build on one another, so perhaps rather multiple methods? (see Feters, Health Services Research 2013) Page 7: "We only include social influences that can be tracked to a single source and incorporate other social influences (such as culture (20), education (21) and perceived discrimination (22)) as potential covariates only." What is meant by Covariates? This is not a statistical analysis, is it? Page 8: how do you know exactly that 2 focus groups and 40 individual interviews will be held? If you do mixed methods, it may appear from the first phase that e.g. more focus groups are needed. And the number of interviews could be based on data saturation so not known beforehand. If the 40 is based on something, please state this. Reference 32 for an Alarming rate is not a good reference, it is a protocol paper (and quite old), please update this. And does this hold for the Netherlands too? Top page 9, this would benefit from a bit more explanation on the choice paradigm, perhaps with an example, since it is central to the study. The first dimension is "the individuals or groups influencing health-related decision-making", but then in the explaining paragraph below it says: "Concerning the first dimension, psychosocial variables are one of the key facets of patient-centred care." But these are not individuals or groups, are they? This is confusing. And "This dimension informs whose preferences or actions must be considered" does that mean considered by the individual who is choosing, or by the framework? Please check language for such inclarities. "Physician Patient interaction" and "social support" are two important constructs in the domain of healthcare. This is vague, what is there relation to choice? The explanation of "the choice constructs that these processes affect" is clear to me re. the first two categories (goals and preferences), but not the 3rd: nonevaluative components of choice models ? (but that comes later, I see). Nevertheless, I do not consider Trade-offs the same as goals, so perhaps some rewording would help in this first construct too. And "alternatives are valued differently based on the goal that a healthcare user seeks to attain vis-à-vis the other persons involved in the decision-making." This is an example that might have been beneficial in one of the more introductory sections above, for naïve readers to have an idea what
--	---

	to think about. And is it always that one chooses because of impacts on others, or could it also be because of e.g. opinions of the others (social norms). And while typing this I realize that it is confusing that above you state that Social Norms are mechanisms, but here social norms could be part of the goals in the construct that these processes affect. Is this not circular? Can it be both mechanism and part of the outcome process, can it? Opinions of others (social norm) is a mechanism influencing, and an outcome in the choice paradigm (goal)? I fear that I am confused now. Hopefully you can rephrase or reorganize a bit to make it more clear. Then: "Preference formation describes the effects of social influences on the acquisition of information about attributes and weights deployed to healthcare users to determine options' total utilities." That is a limited view on preference formation: get the info and there is your preference!? Finally, bottom page 9: non-evaluative components of choice models refer to aspects such as behavioural rules, non-compensatory processes, or screening of alternatives before evaluation . Please help me here a bit too by giving examples. Behavioural rules could well be evaluative, not? Please explain in words what non-evaluative components are before giving these 3 examples. Page 10: a paper cannot have a fulltext available, check language; moreover, I believe Japan is considered Western. Id: "Finally, we will review the initial coding scheme before completing the descriptive analysis." Can you say something more about the analysis if it is something additional ("Finally")? Why the focus groups only on flu vaccination and birth setting? Why not four? If you indeed perform Mixed Methods, the interviews should build on the FG, but you don't have FG for the other two diseases? What is the use of a focus group to test the engagement of participants?! Please explain. Top p 12: before the FG and I patients receive information. Odd location/order of text, I suggest moving this up. And then why a survey/questionnaire (a.o. decision making style) in a qualitative study, this is unusual, or is it to construct table 1? Please explain. Next, the analysis of the FG will focus on the data collection process? What do
--	---

	you mean? Is that relevant for your research questions? And you say you do a partly deductive analysis, then why grounded theory? Grounded theory to me does not seem to fit this study. And do you indeed store data in the archives of the Library? Seems like an odd place. You use experts in the Delphi method, will you also consider Users (clinicians, patients)? Experts “will be proposed parts of the social interdependent choice framework and asked to provide feedback in several rounds”. Please mention that your analysis will lead to this framework (under Analysis). Perhaps also indicating how. Can you please specify in more detail how “Such a framework has the potential to contribute towards more accurate predictions of healthcare choices”? (discussion) Textual: in English it is The Literature, not Literature
--	---

REVIEWER	Forcino, Rachel The University of Kansas School of Medicine, Department of Population Health
REVIEW RETURNED	09-Oct-2023

GENERAL COMMENTS	Thank you for the opportunity to review this protocol for a study developing a novel framework that includes the role of social factors in healthcare decision-making. Minor comments include: Page 6 line 58: “...to extend the existing theoretical choice framework...” Is a reference available for the existing theoretical choice framework that you will build on? Page 9 describes three dimensions of the social interdependent choice paradigm. Is more detail available on how these dimensions were derived, whether by the authors or based on existing theory? It seems like the authors generated these dimensions while taking into account the literature discussed down the page. Same question for the three choice construct categories in page 9 lines 45-50. Page 12 lines 15-16: Are the questions on decision-making style existing, validated measures or bespoke for this study? Clarification questions: Page 8 lines 51-53: does “decrease in active-surveillance compliance” refer to patients who initially choose active surveillance and then later decide to initiate definitive treatment? Page 10 line 25: does “have the full text available” refer to full-text articles available through a specific institutional subscription? Or is it just meant to exclude abstract-only publications, e.g., conference proceedings?
---

	Page 13 line 14: “This will allow the experts to revise their earlier comments.” I’m not sure if I follow what this is describing. Is it asking them to confirm or correct your understanding of their earlier comments that you present to them in the summary, like in member checking? Or is this describing a new Delphi round?
--	--

VERSION 1 – AUTHOR RESPONSE

Reviewer 1

“This is a protocol for an interesting and relevant study on choice models. The aim is to achieve more accurate choice predictions by identifying and integrating into these models the social influences that affect healthcare user decision-making. The authors propose to develop a social interdependent choice framework of healthcare user decision-making. I think this will be an interesting and highly relevant study. My main concerns are textual, upon reading I had many questions, calls for clarification. I provide them simply in the order of the text.”

1. *“A bit confusing was to me that in different places different wording is used, and that at the end, the framework seemed to have disappeared. Moreover how exactly the framework can be used eludes me.”*

Response: We would like to thank the reviewer for their comments. We agree that the link of our proposed work to the final framework can be described more clearly in the manuscript. We have therefore made the following changes:

In the final paragraph of the Delphi section: *“In this case, the version of the socially interdependent choice framework with the most consensus will be accepted, and dissents will be noted.”*

And in the discussion: *“By incorporating these social influences, which can have a big impact on the choice process (8-12), such a framework has the potential to contribute substantially towards more accurate predictions of healthcare choices.”*

The framework can be used by those who investigate and predict multi-attribute decisions in healthcare (scientific researchers) and by those who make decisions in healthcare (e.g., policy makers, care insurance companies, pharmaceutical companies, clinicians). Through this, policy-decisions can be based on accurately predicted healthcare users’ demand. For example, this could lead to less wastage of unused medication. This is more explicitly described now in the introduction and in the discussion.

2. *““Based on the outcomes a social influence independent choice framework” will be drafted (Design section in Abstract). I guess the term social social independent is an error for in the rest of the paper it is interdependent? But perhaps socially interdependent?”*

Response: Indeed, the cited line should read “interdependent”. This has been corrected now in the protocol. We agree that “socially interdependent” makes more sense than “social interdependent”: this has been changed throughout the text.

3. *“I suggest to use the more commonly term Vaccination rather than Immunization.”*

Response: We thank the reviewer for this suggestion. The term vaccination is indeed more common, and we have therefore replaced immunization with vaccination throughout the protocol.

4. *“On page 10, I would have liked to see some more information on the framework of Dellaert, because the framework has such a prominent place in the study, but as a naïve reader I have not gained much idea about what it will look like. Since it is a deductive step, it would help the reader to see this framework. I really like the following sentence about the inductive part.”*

Response: The framework of Dellaert was used in this study to identify the choice constructs that we will use to screen the literature and develop the focus group and interview guides around. Besides using these key-concepts of the model, the structure of the model will not be used in the current study as a basis for the theoretical framework. However, we agree that it is confusing that the framework is cited without any description of it. Therefore, the following sentences have been rewritten/added: *“The creation of a socially interdependent choice framework demands the connection between the first two dimensions and the choice constructs involved in choice econometric models, such as the choice framework of Dellaert et al. (45), which describes how individuals evaluate attributes and make decisions in the presence of multiple goals.”*

5. *“Is it truly mixed methods, or multiple methods? Not much is said about how studies build on one another, so perhaps rather multiple methods? (see Fetters, Health Services Research 2013)”*

Response: The current study is mixed methods because each method is a step towards the completed social interdependent choice framework. First, the literature review will provide relevant concepts from the literature. The interviews will then focus on the outcomes of the literature review, and the results from the interviews will be combined with the literature review to create a preliminary framework. This framework will then be validated using the Delphi method. In these ways, the steps inform the data collection approach of subsequent steps.

6. *“Page 7: “We only include social influences that can be tracked to a single source and incorporate other social influences (such as culture (20), education (21) and perceived discrimination (22)) as potential covariates only.” What is meant by Covariates? This is not a statistical analysis, is it?”*

Response: In the current study, the focus is on social influences that can be tracked to a single source. Other social influences will be documented as they might be of interest for other studies investigating preferences of patients. This sentence has been updated: *“We only include social influences that can be tracked to a single source. Other social influences (such as culture (20), education (21) and perceived discrimination (22)) that cannot be tracked to a single source will be documented only.”*

7. *“Page 8: how do you know exactly that 2 focus groups and 40 individual interviews will be held? If you do mixed methods, it may appear from the first phase that e.g. more focus groups are needed. And the number of interviews could be based on data saturation so not known beforehand. If the 40 is based on something, please state this.”*

Response: 40 interviews in total means 10 interviews per healthcare domain. Generally, 10 interviews are considered a good starting point for interviews that aim to reach data saturation (e. g., Creswell, reference 48 in the paper). In the protocol, it is mentioned that these 10 interviews per healthcare domain are a minimum. We will check whether data saturation has been reached after these 10 and consider conducting more interviews if necessary. The same argument holds for the focus groups, for which 2 is considered a minimum to explore a topic of interest (Krueger & Casey, reference 46 in the paper). We do not aim to reach data saturation with the focus groups as their main aim is to determine if the interview guide is fit for purpose.

8. *“Reference 32 for an Alarming rate is not a good reference, it is a protocol paper (and quite old), please update this. And does this hold for the Netherlands too?”*

Response: The original reference is not a protocol paper, it reports results of the PRIAS “protocol”, in which *protocol* refers to the treatment protocol of active surveillance. The paper itself provides data from the Netherlands. However, the paper has indeed aged. Therefore, a more up-to-date reference (34; from 2018) has been added to the original reference.

9. *“Top page 9, this would benefit from a bit more explanation on the choice paradigm, perhaps with an example, since it is central to the study.”*

Response: The three dimensions concerning the development of the social interdependent choice paradigm are now described in more detail in the other paragraphs on page 9. This first paragraph merely aims to introduce these mechanisms, so that they can be explained further in subsequent paragraphs. These later paragraphs have been adapted to describe the dimensions more clearly based on comments 10 to 17. Changes are described below.

10. *“The first dimension is “the individuals or groups influencing health-related decision-making”, but then in the explaining paragraph below it says: “Concerning the first dimension, psychosocial variables are one of the key facets of patient-centred care.” But these are not individuals or groups, are they? This is confusing. And “This dimension informs whose preferences or actions must be considered” does that mean considered by the individual who is choosing, or by the framework? Please check language for such unclearities.”*

Response: The first dimension concerns the individuals and groups in the decision maker’s social network that have an influence. This refers to which individuals and groups must be included in the framework. To clarify this, the following sentence has been rewritten: *“This first dimension focuses on which individuals’ and groups actions’ must be considered for the development of the theoretical framework for healthcare user decision-making.”*

11. *““Physician Patient interaction” and “social support” are two important constructs in the domain of healthcare. This is vague, what is their relation to choice?”*

Response: Physician-patient interaction and social support are examples of social influence mechanisms that may impact healthcare user decision-making. To clarify, the respective sentence was rewritten: *“Physician-patient interaction (36-37) and social support (40) are two relevant and well-defined constructs that may impact healthcare user decision-making.”* How exactly these concepts relate to decision-making will be explored in the study.

12. *“The explanation of “the choice constructs that these processes affect” is clear to me re. the first two categories (goals and preferences), but not the 3rd: non-evaluative components of choice models? (but that comes later, I see).”*

Response: Indeed, this is discussed later in the text. To clarify this more, some changes have been made. See our response to comment 17.

13. *“Nevertheless, I do not consider Trade-offs the same as goals, so perhaps some rewording would help in this first construct too.”*

Response: Trade-offs and goals are not viewed as the same construct. However, healthcare users can have different goals regarding choice of treatment. This can then impact the personal trade-offs in characteristics of treatments the healthcare user is willing to make.

14. *“And “alternatives are valued differently based on the goal that a healthcare user seeks to attain vis-à-vis the other persons involved in the decision-making.” This is an example that might have been beneficial in one of the more introductory sections above, for naïve readers to have an idea what to think about.”*

Response: As described at review comment 9, this paragraph serves to give a more in-depth explanation of what choice constructs constitute in the current study. The more introductory section at the top of page 9 merely mentions the three dimensions that are discussed in subsequent paragraphs (including choice constructs in this paragraph). So readers can already form an idea of what choice constructs refer to after reading this paragraph.

15. *“And is it always that one chooses because of impacts on others, or could it also be because of e.g. opinions of the others (social norms). And while typing this I realize that it is confusing that above you state that Social Norms are mechanisms, but here social norms could be part of the goals in the construct that these processes affect. Is this not circular? Can it be both mechanism and part of the outcome process, can it? Opinions of others (social norm) is a mechanism influencing, and an outcome in the choice paradigm (goal)? I fear that I am confused now. Hopefully you can rephrase or reorganize a bit to make it more clear.”*

Response: This section describes the three choice constructs, and some examples are given of how these constructs could be socially influenced. In the case of goals (the first choice construct), other persons may influence the activation of certain goals. This can happen through different social influence mechanisms, one of which is social norms. For example, healthcare users may perceive a social norm to get vaccinated with the goal of protecting others. The goal is then not the social norm, but the goal is protecting others. To clarify the phrasing, the following sentence has been rewritten: *“...where different alternatives are valued differently based on the goal that a healthcare user seeks to attain, with other persons possibly influencing the activation of certain goals.”*

16. *“Then: “Preference formation describes the effects of social influences on the acquisition of information about attributes and weights deployed to healthcare users to determine options’ total utilities.” That is a limited view on preference formation: get the info and there is your preference!?”*

Response: This sentence reads like *information* refers to both *attributes* and *weights*. However, *acquisition* is supposed to refer to *information about attributes*, and to *weights*. Thus, weights do not rely only on the acquisition of information. The sentence has been changed to better reflect this: *“Preference formation describes the effects of social influence on the acquisition of information about attributes and the deployment of weights to these attributes to determine options’ total utilities.”*

17. *“Finally, bottom page 9: non-evaluative components of choice models refer to aspects such as behavioural rules, non-compensatory processes, or screening of alternatives before evaluation. Please help me here a bit too by giving examples. Behavioural rules could well be evaluative, not? Please explain in words what non-evaluative components are before giving these 3 examples.”*

Response: The cited sentence has been split into two, where the first sentence provides an explanation, and the second sentence gives examples. Additionally, the term *behavioural rules* is replaced by a more specific example, *habitual behaviour*. “*Finally, non-evaluative components refer to any steps of the decision-making process where certain options are excluded based on single attributes, instead of an option’s unique blend of attribute levels. Examples of non-evaluative components are habitual behaviour, non-compensatory processes, or screening of alternatives before evaluation.*”

18. “Page 10: a paper cannot have a fulltext available, check language.”

Response: The sentence in question was altered: “*The inclusion criteria further require that the full text of a paper is accessible, ...*”

19. “I believe Japan is considered Western.”

Response: There is no consensus on whether Japan is or is not Western. The literature review will only include countries that are always considered Western and does not include Japan nor any South American countries, which are also sometimes regarded as Western, but other times they are not.

20. “Id: “Finally, we will review the initial coding scheme before completing the descriptive analysis.” Can you say something more about the analysis if it is something additional (“Finally”)?”

Response: The first steps are described earlier in this paragraph. This final sentence is indeed worded a bit strangely. It was removed and replaced with: “*The last step of the process is to describe frequencies and co-occurrences of relevant individuals and groups, social influence mechanisms and choice constructs, and differences across medical areas.*”

21. “Why the focus groups only on flu vaccination and birth setting? Why not four? If you indeed perform Mixed Methods, the interviews should build on the FG, but you don’t have FG for the other two diseases?”

Response: Questions asked in the focus groups and interviews will require participants to have a basic understanding of the different choice constructs we refer to. It is quite challenging to articulate questions in such a way that participants will understand what we are referring to. To ensure the interview guides are fit for purpose, focus groups are organized. These focus groups allow us to test our questions with multiple participants in a short time. We anticipate that only two focus groups will provide us with initial insights on the suitability of the interview guide. Of course, focus groups also provide information related to the research question, so they will be analysed in this way as well, but this is not the main reason for hosting the focus groups. The complex choice constructs are discussed similarly across all four domains, thus we expect two focus groups to be enough to assess the clarity of the interview guide for all four domains.

22. “What is the use of a focus group to test the engagement of participants?! Please explain.”

Response: The response to comment 21 clarifies this more. However, *engagement* is indeed not the right word here. It has been replaced with *understanding*.

23. *“Top p 12: before the FG and I patients receive information. Odd location/order of text, I suggest moving this up. And then why a survey/questionnaire (a.o. decision making style) in a qualitative study, this is unusual, or is it to construct table 1? Please explain.”*

Response: There are two reasons for inclusion of a survey. First, it is to gather some demographic information and to make sure we have a sufficiently heterogeneous sample. Second, quantitative data on decision-making style will allow us to contextualize the data and perhaps explain some differences in social influence based on decision-making style.

Within health economics, it is not unusual to include tasks in interviews and focus groups alongside collecting qualitative data, to focus on what are otherwise quite abstract concepts for people (e.g., Canaway A, Al-Janabi H, Kinghorn P et al. Close-person spill-overs in end-of-life care: using hierarchical mapping to identify whose outcomes to include in economic evaluations. *PharmacoEconomics*. 2019;37:573-583. <https://doi.org/10.1007/s40273-019-00786-5> and Coast J, Bailey C, Canaway A, Kinghorn P. “It is not a scientific number it is just a feeling”: Populating a multi-dimensional end-of-life decision framework using deliberative methods. *Health Economics*. 2021;30(1):1033-1049. <http://dx.doi.org/10.1002/hec.4239>).

Regarding the placement of this section, it follows a description of the focus groups and interviews, because the survey is regarded as an ‘extra’ to help contextualize the interview findings. It is thus not as important as the interviews, which is why it is described more briefly and later in the text.

24. *“Next, the analysis of the FG will focus on the data collection process? What do you mean? Is that relevant for your research questions? And you say you do a partly deductive analysis, then why grounded theory? Grounded theory to me does not seem to fit this study.”*

Response: The analysis of the focus groups has two goals, as described in our response to comment 21. Analyzing the clarity of the interview guide is not directly related to the research questions but will contribute to refining the questions for the interviews. The focus groups and interviews are mainly analyzed inductively, as the aim theory generation. The use of constant comparative analytical approaches alongside the use of an economic lens in analysing data is described in Coast J, Jackson L. Understanding primary data analysis. In: Coast J, editor. *Qualitative methods for health economics*. Rowman & Littlefield International; 2017. Additionally, the literature review may provide some important social influences, mechanisms, or choice constructs from the literature. These will be deductively utilized in the analysis of the interviews, in the sense that they are treated as sensitizing concepts during the qualitative analysis: the researchers will keep these constructs in mind to help contextualize information that participants provide. The use of sensitizing concepts in this way is described in detail by: Bowen GA. Grounded Theory and Sensitizing Concepts. *International Journal of Qualitative Methods*. 2006;5(3):12-23. Available from: <https://doi.org/10.1177/160940690600500304>

25. *“And do you indeed store data in the archives of the Library? Seems like an odd place.”*

Response: Recent policy changes at university now require us to archive our data through a dedicated application called SURF Yoda. The protocol has been updated to reflect this.

26. *“You use experts in the Delphi method, will you also consider Users (clinicians, patients)?”*

Response: Clinicians are also regarded as experts in the current study. This is now clarified in brackets: *“Experts (researchers and clinicians) from the different fields...”*. Clinicians will

also be the users of the eventual social interdependent choice framework. Patients provided significant input for the development of the framework during the interview and focus group phases. Since they will not apply the framework in choice modelling (unless they are researchers or clinicians, and via that capacity included in the Delphi study) we feel they are not the end-users of the framework. Additionally, building a framework requires a particular set of skills and expertise.

27. *“Experts “will be proposed parts of the social interdependent choice framework and asked to provide feedback in several rounds”. Please mention that your analysis will lead to this framework (under Analysis). Perhaps also indicating how.”*

Response: The following sentence has been added: *“The analysis will result in a first version of the theoretical framework of social influence on healthcare user decision-making.”* This will be achieved through the constant comparison approach described earlier in this section.

28. *“Can you please specify in more detail how “Such a framework has the potential to contribute towards more accurate predictions of healthcare choices”? (discussion)”*

Response: Prior research has shown that choices in the healthcare domain can be (entirely) socially determined. Incorporating these social influences into the framework, would thus lead to better predictions of healthcare users’ choices. The basic logic here is that better characterization of patients’ preferences will naturally lead to better prediction of choice behaviour. To clarify this, the following line has been added to the discussion section: *“By incorporating these social influences, which can have a big impact on the choice process (8-12), such a framework has the potential to contribute towards more accurate predictions of healthcare choices, which could in turn have a positive impact on decreasing poor policy decisions, trial-and-error implementations, and demand-supply imbalances.”*

29. *“Textual: in English it is The Literature, not Literature.”*

Response: The word the has been added to the first sentence of the Delphi method section.

Reviewer 2

“Thank you for the opportunity to review this protocol for a study developing a novel framework that includes the role of social factors in healthcare decision-making.”

1. *“Page 6 line 58: “...to extend the existing theoretical choice framework...” Is a reference available for the existing theoretical choice framework that you will build on?”*

Response: We thank the reviewer for their comments. The existing theoretical choice framework does not refer to a specific work, but rather our current understanding of decision-making in general. Later in the paper, we refer to the work of Dellaert et al. (45) because of the choice constructs that Dellaert’s work identifies, that we will use to screen the literature and develop the focus group and interview guides around. We have decided not to refer to Dellaert in the quoted line, to avoid any misunderstanding that we are building directly on Dellaert’s framework, while rather we are constructing an entirely new framework. Still, Dellaert is referenced later in the text.

2. *“Page 9 describes three dimensions of the social interdependent choice paradigm. Is more detail available on how these dimensions were derived, whether by the authors or based on*

existing theory? It seems like the authors generated these dimensions while taking into account the literature discussed down the page. Same question for the three choice construct categories in page 9 lines 45-50.”

Response: Indeed, these dimensions were taken from literature. Heany & Israel (reference 40 in the paper) build on work by House et al. (reference 38 in the paper) and discuss social influence in terms of who provides what (to whom). This contains the first two dimensions: the individuals or groups influencing the decision-making, and the mechanisms through which this influence affects the decision-making. The final dimension concerns the choice constructs as described by Dellaert et al. (45). References to these three papers have been added to their respective paragraphs.

3. *“Page 12 lines 15-16: Are the questions on decision-making style existing, validated measures or bespoke for this study?”*

Response: Existing, validated measures will be used. Decision-making style will be measured using a validated questionnaire developed by Pachur & Spaar (reference 49 in paper). Additionally, health literacy will be measured using a validated questionnaire developed by Chew et al. (reference 50 in paper). These references have been added to the paper.

4. *“Page 8 lines 51-53: does “decrease in active-surveillance compliance” refer to patients who initially choose active surveillance and then later decide to initiate definitive treatment?”*

Response: A decrease in active-surveillance compliance refers to patients who choose active surveillance and at a later point stop showing up for the check-up appointments part of the active surveillance program. Patients who initially choose active surveillance and then later have to switch to active treatment due to tumor progression are not regarded as dropouts, as this is considered part of the natural trajectory of the active surveillance program.

5. *“Page 10 line 25: does “have the full text available” refer to full-text articles available through a specific institutional subscription? Or is it just meant to exclude abstract-only publications, e.g., conference proceedings?”*

Response: This refers to the full text publications available through the EUR institutional subscription. Papers that were not accessible through this subscription were excluded from the literature review.

6. *“Page 13 line 14: “This will allow the experts to revise their earlier comments.” I’m not sure if I follow what this is describing. Is it asking them to confirm or correct your understanding of their earlier comments that you present to them in the summary, like in member checking? Or is this describing a new Delphi round?”*

Response: This section is describing a new Delphi round. In a new round, there will be some new questions for the participants, in addition to a summary of the comments we received in the previous round. Participants are asked to comment on the new questions, but may also revise their answers from the previous round based on the summary.

VERSION 2 – REVIEW

REVIEWER	Forcino, Rachel The University of Kansas School of Medicine, Department of Population Health
REVIEW RETURNED	11-Jan-2024

GENERAL COMMENTS	The authors have provided detailed responses to my previous comments. I found the responses very helpful in clarifying what were ambiguous and/or unclear aspects of the original protocol submission; however, many are not reflected in the manuscript. Those points should be addressed in the manuscript so that other readers may benefit from the authors' helpful explanations and clarifications. Among the other points of clarification, details of the systematic review inclusion criteria (i.e. inclusion based on a specific subscription) should be specified in text. Minor comment: "After consent has been obtained, they will be asked to fill in a short (online) survey including questions on their health literacy (49) and their decision-making style (50)." References 49 (Pachur) and 50 (Chew) should be switched here.
---

VERSION 2 – AUTHOR RESPONSE

Reviewer 2

"The authors have provided detailed responses to my previous comments. I found the responses very helpful in clarifying what were ambiguous and/or unclear aspects of the original protocol submission; however, many are not reflected in the manuscript. Those points should be addressed in the manuscript so that other readers may benefit from the authors' helpful explanations and clarifications. Among the other points of clarification, details of the systematic review inclusion criteria (i.e. inclusion based on a specific subscription) should be specified in text."

Response: We thank the reviewer for their comments. We have revised the protocol based on three reviewer comments that we indeed explained and clarified in our previous response letter but were not (sufficiently) reflected in the previous resubmitted manuscript. The other three comments were already addressed in the previously resubmitted protocol. Clarification for the changes that we made in this new version was provided in the previous response letter. The reviewer comments and changes are as follows:

- *"Page 8 lines 51-53: does "decrease in active-surveillance compliance" refer to patients who initially choose active surveillance and then later decide to initiate definitive treatment?"*

Response: We have rephrased: *"there is an alarming increase in active-surveillance drop-out from prostate cancer patients (32-33)."*

- *"Page 10 line 25: does "have the full text available" refer to full-text articles available through a specific institutional subscription? Or is it just meant to exclude abstract-only publications, e.g., conference proceedings?"*

Response: We have rephrased: *“The inclusion criteria further require that the full text of a paper is accessible through the institutional subscription of any of the project collaborators, ...”*

- *“Page 13 line 14: “This will allow the experts to revise their earlier comments.” I’m not sure if I follow what this is describing. Is it asking them to confirm or correct your understanding of their earlier comments that you present to them in the summary, like in member checking? Or is this describing a new Delphi round?”*

Response: We have added: *“This will allow the experts to revise their earlier comments in the next round of the Delphi study.”*

*“After consent has been obtained, they will be asked to fill in a short (online) survey including questions on their health literacy (49) and their decision-making style (50).’
References 49 (Pachur) and 50 (Chew) should be switched here.”*

Response: We thank the reviewer for the correction. We have switched the references in the reference list: Chew (49) and Pachur (50).